# The semiquinone swing in the bifurcating electron transferring flavoprotein/butyryl-CoA dehydrogenase complex from *Clostridium difficile*

Julius K. Demmer[1], Nilanjan Pal Chowdhury[2,3], Thorsten Selmer[4], Ulrich Ermler[1] & Wolfgang Buckel [ID] [2,3]

The electron transferring flavoprotein/butyryl-CoA dehydrogenase (EtfAB/Bcd) catalyzes the reduction of one crotonyl-CoA and two ferredoxins by two NADH within a flavin-based electron-bifurcating process. Here we report on the X-ray structure of the *Clostridium difficile* (EtfAB/Bcd)$_4$ complex in the dehydrogenase-conducting D-state, α-FAD (bound to domain II of EtfA) and δ-FAD (bound to Bcd) being 8 Å apart. Superimposing *Acidaminococcus fermentans* EtfAB onto *C. difficile* EtfAB/Bcd reveals a rotation of domain II of nearly 80°. Further rotation by 10° brings EtfAB into the bifurcating B-state, α-FAD and β-FAD (bound to EtfB) being 14 Å apart. This dual binding mode of domain II, substantiated by mutational studies, resembles findings in non-bifurcating EtfAB/acyl-CoA dehydrogenase complexes. In our proposed mechanism, NADH reduces β-FAD, which bifurcates. One electron goes to ferredoxin and one to α-FAD, which swings over to reduce δ-FAD to the semiquinone. Repetition affords a second reduced ferredoxin and δ-FADH$^-$, which reduces crotonyl-CoA.

[1] Max-Planck-Institut für Biophysik, Max-von-Laue-Str. 3, 60438 Frankfurt am Main, Germany. [2] Laboratorium für Mikrobiologie, Fachbereich Biologie and SYNMIKRO, Philipps-Universität, 35032 Marburg, Germany. [3] Max-Planck-Institut für terrestrische Mikrobiologie, Karl-von-Frisch-Str. 10, 35043 Marburg, Germany. [4] Fachbereich Chemie und Biotechnologie, FH Aachen, Heinrich-Mußmann-Str. 1, 52428 Jülich, Germany. Julius K. Demmer and Nilanjan Pal Chowdhury contributed equally to this work. Correspondence and requests for materials should be addressed to U.E. (email: ulrich.ermler@biophys.mpg.de) or to W.B. (email: buckel@staff.uni-marburg.de)

Flavin-based electron bifurcation (FBEB) has been recognized as a new mode of energy coupling in strictly anaerobic bacteria and archaea[1–4]. In the applied enzyme machineries, a flavin hydroquinone (HQ or FH⁻) acts as the bifurcating donor; one electron flows to a high-potential acceptor, whereas the other electron reduces the low potential ferredoxin or flavodoxin. Thus, the exergonic reduction of such an acceptor drives the endergonic electron transfer (ET) to ferredoxin or flavodoxin; in other words: an exergonic reaction is coupled to an endergonic reaction. Reduced ferredoxin and flavodoxin can be regarded as "energy rich" compounds, which in many fermenting anaerobes form $H_2$ and generate an electrochemical $Na^+$ gradient for ATP synthesis via ferredoxin-$NAD^+$ reductase (Rnf)[5, 6]. In addition, they reduce $CO_2$ to CO in acetogens[7, 8] and to a formyl group in methanogens[9, 10]. Flavins, FMN or FAD, are suitable coupling agents as they can adopt a two-electron redox state, quinone/hydroquinone (Q/HQ or FAD/FADH⁻) as well as two one-electron redox states, Q/SQ and SQ/HQ (SQ = semiquinone, FAD•⁻), one with a strong and a second with a weak electron donor potential and vice versa. An analogous process known as Q-cycle was reported for the membrane-spanning cytochrome $bc_1$ complex where ubihydroquinone is the site of quinone-based electron bifurcation[11, 12].

Meanwhile seven FBEB enzymes are established[13–18]. Besides NADH-dependent ferredoxin: $NADP^+$ oxidoreductase (Nfn)[8, 17, 19–21], the best studied FBEB system is the electron transferring flavoprotein/butyryl-CoA dehydrogenase (EtfAB/Bcd) complex, which is a key enzyme in butyrate producing anaerobes[1–3, 9, 13]. Various biochemical, kinetic, spectroscopic, structural and mechanistic data are already reported[6, 13, 22–24]. The EtfAB/Bcd complex is composed of subunits EtfA, EtfB and Bcd; each subunit contains one FAD as prosthetic group, termed α-FAD, β-FAD and δ-FAD, respectively. The electron-bifurcating EtfAB/Bcd complex couples the exergonic reduction of crotonyl-CoA to butyryl-CoA ($E_0' = -10$ mV)[25] with the endergonic reduction of 2 ferredoxins ($E_0' = -405$ mV)[26] by oxidation of 2 NADH to 2 $NAD^+$ ($E_0' = -320$ mV) (Fig. 1). According to current knowledge[6, 23, 24], a hydride is transferred from NADH to the bifurcating β-FAD. The generated β-FADH⁻ donates one electron to α-FAD, most likely the stable anionic semiquinone α-FAD•⁻, yielding α-FADH⁻, and the other electron to one [4Fe-4S] cluster

of ferredoxin. After domain rearrangement as shown in this work, one electron flows from α-FADH⁻ to δ-FAD forming presumably δ-FADH•. Repetition of this process affords a second reduced ferredoxin and the returned α-FAD•⁻ becomes reduced to α-FADH⁻, which donates one electron to δ-FADH•. Finally, the generated δ-FADH⁻ transfers a hydride to crotonyl-CoA yielding butyryl-CoA. Ferredoxin can be equally well replaced by flavodoxin[6] and even by $O_2$[13], which is reduced to superoxide $O_2•⁻$[23].

There are two types of bifurcating EtfAB-Bcd complexes. In the Gram-negative Firmicutes, *Acidaminococcus fermentans*[24] and *Megasphaera elsdenii*[23], EtfAB and the tetrameric (Bcd)₄ are separate proteins, which form an (EtfAB)₂-(Bcd)₄ complex during catalysis[24]. In contrast, the complexes present in the Gram-positive Clostridia, a main class of the Firmicutes, do not dissociate during purification. Examples are the tight complexes from *Clostridium kluyveri*[2], *Clostridium tetanomorphum*[27] and *Clostridium difficile*[13], the latter studied in this work. Recently, the crystal structure of EtfAB from *A. fermentans* has been solved[24]. It closely resembles that of the non-bifurcating EtfABs from aerobes and respiring anaerobes but contains β-FAD at the place of AMP in addition to the α-FAD[28–30]. All EtfABs are composed of three domains; domains I and II are components of the A- or α-subunit, and domain III of the B- or β-subunit. α-FAD is located in the mobile domain II and β-FAD sits between the rigid domains I and III. The distance between α-FAD and β-FAD amounts to 18 Å, which can be reduced to 14 Å by moving domain II towards domains I + III without causing serious steric clashes[24]. Soaking the EtfAB crystal with NADH revealed a binding close to β-FAD, though only the ADP part of NADH could be structurally resolved. Since the known structures of EtfAB and Bcd from *M. elsdenii*[31] and *A. fermentans*[24] cannot tell how EtfAB is bound relative to Bcd, we solved the structure of the *C. difficile* complex as shown in this work. Surprisingly, in this structure the position of α-FAD is close to δ-FAD of one Bcd subunit (8.5 Å distance) ready for electron donation (D state; 37 Å distance between α-FAD and β-FAD). By combining the structures of *A. fermentans* Etf and *C. difficile* EtfAB/Bcd, we built a FBEB-like state of the whole complex (B-like state; 18 Å distance between α-FAD and β-FAD) and modeled the FBEB state (B state; 14 Å distance between α-FAD and β-FAD). We also explored conformational changes by site-directed mutagenesis

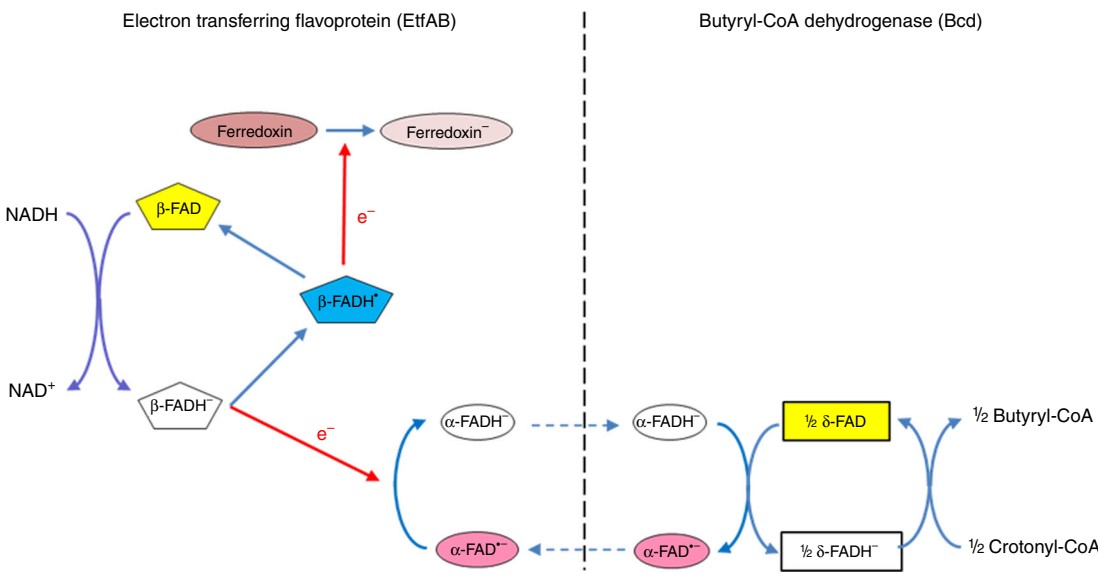

**Fig. 1** Reaction of the EtfAB/Bcd complex. In the first round the EtfAB/Bcd complex catalyzes the NADH-dependent reduction of β-FAD to β-FADH⁻, which bifurcates. One electron reduces α-FAD•⁻ to α-FADH⁻ and the other ferredoxin to ferredoxin⁻. α-FADH⁻ swings to δ-FAD forming δ-FADH• and α-FAD•⁻, which returns to β-FAD. The second round generates a second ferredoxin⁻ and δ-FADH⁻, which reduces crotonyl-CoA to butyryl-CoA

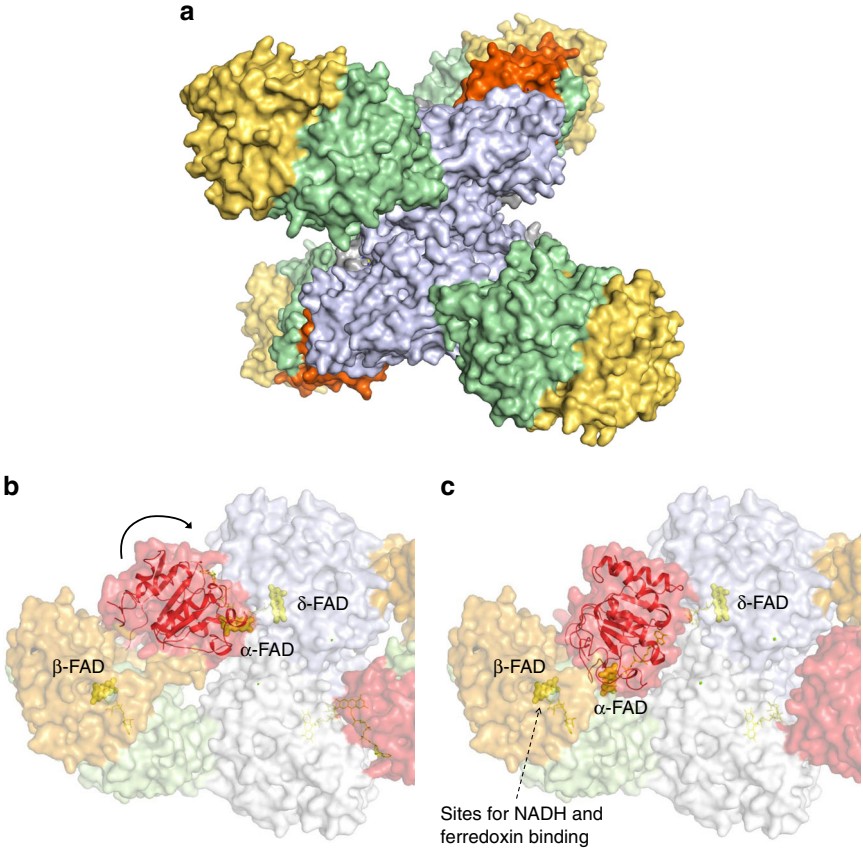

**Fig. 2** Structure of the EtfAB/Bcd complex. **a** The heterododecameric (EtfAB/Bcd)$_4$ complex. Tetrameric Bcd (grey) forms the core and the four EtfAB units are peripherally associated (domain I in brown, domain II in red and domain III in green). **b** EtfAB/Bcd of *C. difficile* in the D state. The catalytic reaction involves domain I (brown) of EtfA and domain III of EtfB (green) forming the EtfAB base and a Bcd dimer (Bcd1 in grey and Bcd2 in light blue). Each of the three subunits Bcd, EtfA and EtfB non-covalently binds one FAD (isoalloxazine in yellow). **c** The EtfAB/Bcd complex in the B-like state. Domain II is rotated nearly 80° such that α-FAD and β-FAD are nearly in a distance suitable for ET. α-FAD embedded into the weakly associated domain II serves as a one-electron shuttle between the electron-donating β-FADH⁻ and the electron-accepting δ-FAD

and found that the α-FAD on domain II acts like an electron seesaw swinging between the B and D states.

## Results

**Overall structure of the EtfAB/Bcd complex.** The EtfAB/Bcd complex was found in a heterododecameric (EtfAB/Bcd)$_4$ state with a molecular mass of ca. 440 kDa (Fig. 2). Architecturally, the tetrameric Bcd forms the core of the enzyme complex to which four EtfAB units are peripherally attached. The four EtfAB are spatially separated suggesting four independent catalytic processes. In comparison, the related non-bifurcating EtfAB/dehydrogenases were structurally characterized as EtfAB/(MCAD)$_4$[32] and (EtfAB)$_2$/(TMADH)$_2$ complexes[30].

As described previously[24, 28, 30], EtfAB is built up of three domains, of which I (1–199) and II (200–331) belong to subunit A and III (5–265) to subunit B. The structurally related domains I and III are tightly associated (Supplementary Fig. 1) and constitute the EtfAB base. In contrast to domain I, domain III contains an exposed segment after strand 2:7, termed EtfB protrusion. Domain II of EtfA is composed of a flavodoxin-like fold enlarged by a sixth parallel strand and a helix from the C-terminal extension of EtfB, referred to as the EtfB arm (235–265). Each Bcd monomer is built up of an N-terminal helical domain (1–116), a medial β-sheet domain (117–227) and a C-terminal helical domain (228–378) (Supplementary Fig. 1).

Each EtfAB unit forms two contact areas with a Bcd dimer (the monomers named Bcd1 and Bcd2) that are placed around 50 Å

apart from each other (Fig. 2). The first contact area is located between the N-terminal domain of Bcd1 and the EtfAB base and is considered as fixed during the reaction cycle. An extended hydrophobic interface is formed by helices 5:23 and 39:47 of Bcd1 and helices 60:66 and 191:201 (and the preceding segment) of EtfB (Supplementary Fig. 1).

The second contact region is formed between domain II and the Bcd1–Bcd2 dimer (Fig. 2, Supplementary Fig. 1). The β-sheet domain of Bcd2 and loops connecting helices of the C-terminal domain of Bcd1 faces the C-terminal loops of the central β-sheet of domain II. The interface is primarily characterized by a small hydrophobic patch around α-FAD and by a few long interacting side chains (frequently charged), i.e., E198/D345 of Bcd1/ Bcd2 and R243 of domain II and E210 of Bcd1 and R127 of domain II (Fig. 3).

In the determined EtfAB/Bcd structure, the EtfAB base and domain II forms no real interface; the highly flexible C-terminal ends of EtfB and EtfA (B > 150 Å²) touching Bcd only form negligible interactions. The covalent linkers between domains I and II and between domain III and the EtfB arm (Supplementary Fig. 1) are completely solvent-accessible and bendable. Therefore, domain II is kept in its position by its interactions with Bcd.

A superposition of the separate EtfAB and Bcd of *A. fermentans*[24] onto the EtfAB/Bcd complex of *C. difficile* revealed domains I, III and Bcd well aligned while the two domains II are positioned in completely different orientations. The rotation angle between domains II is 78° (center shift: 5 Å) and the rmsd is

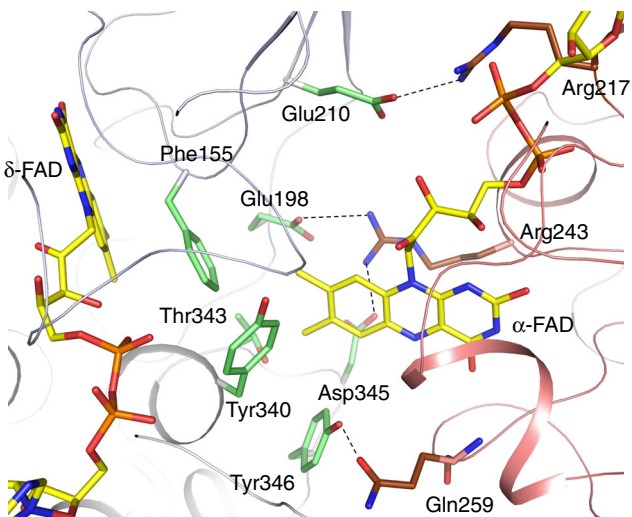

**Fig. 3** Coupling site between α-FAD and δ-FAD in the D state. The isoalloxazine rings are oriented perpendicularly to each other with the xylene ring of α-FAD pointing towards the *Si*-side of the isoalloxazine ring of the δ-FAD. The binding sites for β-FAD and δ-FAD are virtually identical in D and B-like states; substantial differences exists for α-FAD. The exposed xylene ring of α-FAD is flanked by predominantly hydrophobic side chains (carbon in green) from the Bcd dimer (white blue, grey). Phe155 of Bcd1 is in van der Waals contact to both isoalloxazine rings. The shortest distance between α-FAD and δ-FAD is between the C8 methyl groups. The ADP and ribitol moieties of α-FAD are surface-exposed and only contacted to Bcd via a layer of solvent molecules. The rather weak interaction between Bcd and α-FAD supports a quick detachment process required for catalysis

1.3 Å essentially indicating a rigid-body movement (Fig. 2b, c). Larger deviations are only detectable in the linking segments and the flexible C-terminal helices of EtfA and EtfB. The reorientation of domain II, of course, also results in a different assembly between domain II and the rest protein. Domain II only forms a small interface to Bcd mainly involving the EtfB arm but a larger interface to the EtfAB base (Fig. 2). Except for the two flexible covalent linkers the contact is predominantly mediated via a layer of solvent molecules[24].

**Electron transfer between α-FAD and β-FAD and δ-FAD.** The different orientation of domain II in the EtfAB/Bcd complex implicates different distances between α-FAD, β-FAD and δ-FAD. In the determined EtfAB/Bcd structure characterized by a pronounced domain II-Bcd dimer interface, the distance between α-FAD and δ-FAD is 8.4 Å thereby indicating that the EtfAB/Bcd complex is trapped in a dehydrogenase-conducting conformation termed D state (Fig. 2b). The edge-to-edge distance between the β-FAD and α-FAD isoalloxazines is 37 Å which certainly excludes mutual ET.

In the EtfAB/Bcd complex generated by superimposing the separated EtfAB and Bcd structures of *A. fermentans* on the EtfAB/Bcd complex of *C. difficile* in the D state, the isoalloxazine rings of α-FAD and δ-FAD have an edge-to-edge distance of 35 Å which blocks mutual ET. In contrast, the distance between α-FAD and β-FAD is 18 Å that appears to be slightly too long for an efficient ET[33]. This EtfAB/Bcd complex conformation is therefore named as the FBEB conducting-like or B-like state (Fig. 2c). Notably, the EtfB arm in the B-like state slightly overlaps with Bcd suggesting that this state represents an artificial conformation only realized in the absence of Bcd. Due to the described soft solvent-mediated domain II—EtfAB base interface, a further rotation of domain II without severe steric clashes is feasible by

which the distance between the isoalloxazine rings of α-FAD and β-FAD is reduced to ca. 14 Å (Fig. 4). This conformation of the EtfAB/Bcd complex is therefore named as the FBEB conducting or B state.

**Polypeptide surrounding of the FADs.** In all three EtfAB/Bcd states, the δ-FAD isoalloxazine ring is completely shielded from the respective domain II-Bcd interfaces (Fig. 3) such that its redox potential is fairly independent of the domain II orientation. While the substrate crotonyl-CoA binds at the *Re*-side of δ-FAD, the F155 benzyl group of Bcd1 at the *Si*-side bridges δ-FAD and α-FAD in the D state. In the B-like state the EtfB arm contacts Bcd (Fig. 2b).

In the D state, α-FAD is involved in the domain II-Bcd interface and the exposed xylene ring is completely encapsulated by T343, Y346, and Y340 of Bcd2 as well as F155 and E198 of Bcd1. This hydrophobic patch significantly contributes to the domain II - Bcd interface (Fig. 3). The reorientation of domain II into the B-like state results in substantial changes of the environment of α-FAD. Now, the xylene group is contacted by P36, Y37 and Y40 of the EtfAB base but remains largely solvent-exposed[24]. Interestingly, the different interface partner also causes a rotation of ca. 15° between the isoalloxazine rings of the D and B-like states. The interactions between α-FAD and domain II are essentially independent on the state and, in particular, characterized by the shielding of N5 by S270 and a surrounding of the N1-C2 = O group suitable for the stabilization of α-FAD[•−] and α-FADH[−]. Altogether, the α-FAD surrounding at the domain II-Bcd interface of the D state is more hydrophobic than that at the domain II—EtfAB base interface in the B-like state.

The binding mode of the bifurcating β-FAD appears to be virtually identical in the D and B-like states (Fig. 2). The solvent-exposed isoalloxazine ring projected from domain III towards domain I is partly shielded by a hairpin-like segment of domain I termed EtfA hairpin and the anchor of the EtfB arm which form together a three-stranded β-sheet (Fig. 4). The interactions to the isoalloxazine ring are strictly conserved including those between EtfA R136 and N5 (Supplementary Fig. 2) which stabilize the oxidized state and thus a relatively low redox potential of the β-FAD[•−]/β-FAD pair. A ferredoxin can be attached adjacent to the solvent-exposed isoalloxazine ring with an edge-to-edge distance between its [4Fe-4S] cluster and β-FAD of ca. 8 Å independent on the presence of Bcd (Supplementary Fig. 3)[24].

**Site-specific mutations in the EtfAB/Bcd complex.** The EtfAB/Bcd variants of *C. difficile* (as the wildtype) were produced in *Escherichia coli* and purified to homogeneity with yields varying between 2 to 20 mg from 4 L cultures. The variants were analyzed by measuring kinetic activities with respect to FBEB, NADH oxidase, and butyryl-CoA dehydrogenase (Fig. 5, Supplementary Table 1 and Supplementary Fig. 4). To explore the rearrangement of domain II between two electron-conducting sites in solution, we site-specifically exchanged amino acids in the domain II—Bcd and domain II—EtfAB base interfaces (Fig. 2). EtfA R243, Bcd1 E198 and Bcd2 D345 form ionic interactions in the D state (Fig. 3). Due to their high conservation and stimulated by studies on the non-bifurcating human EtfAB/MCAD complex[32], these residues were mutated. As expected the activity of the resulting variants, measured with the three assays, were drastically reduced (Fig. 5). EtfB I125 is located at the soft domain II—EtfAB base interface. The low enzymatic activity of the I125D/F variant suggests that the rotation of domain II between the D and B states is hampered. EtfA R255 and EtfB F233 are part of the fixation of the covalent linker between domains I and II and between domain III and the EtfB arm. Their exchange to glutamine and aspartate substantially increases the catalytic activities, which

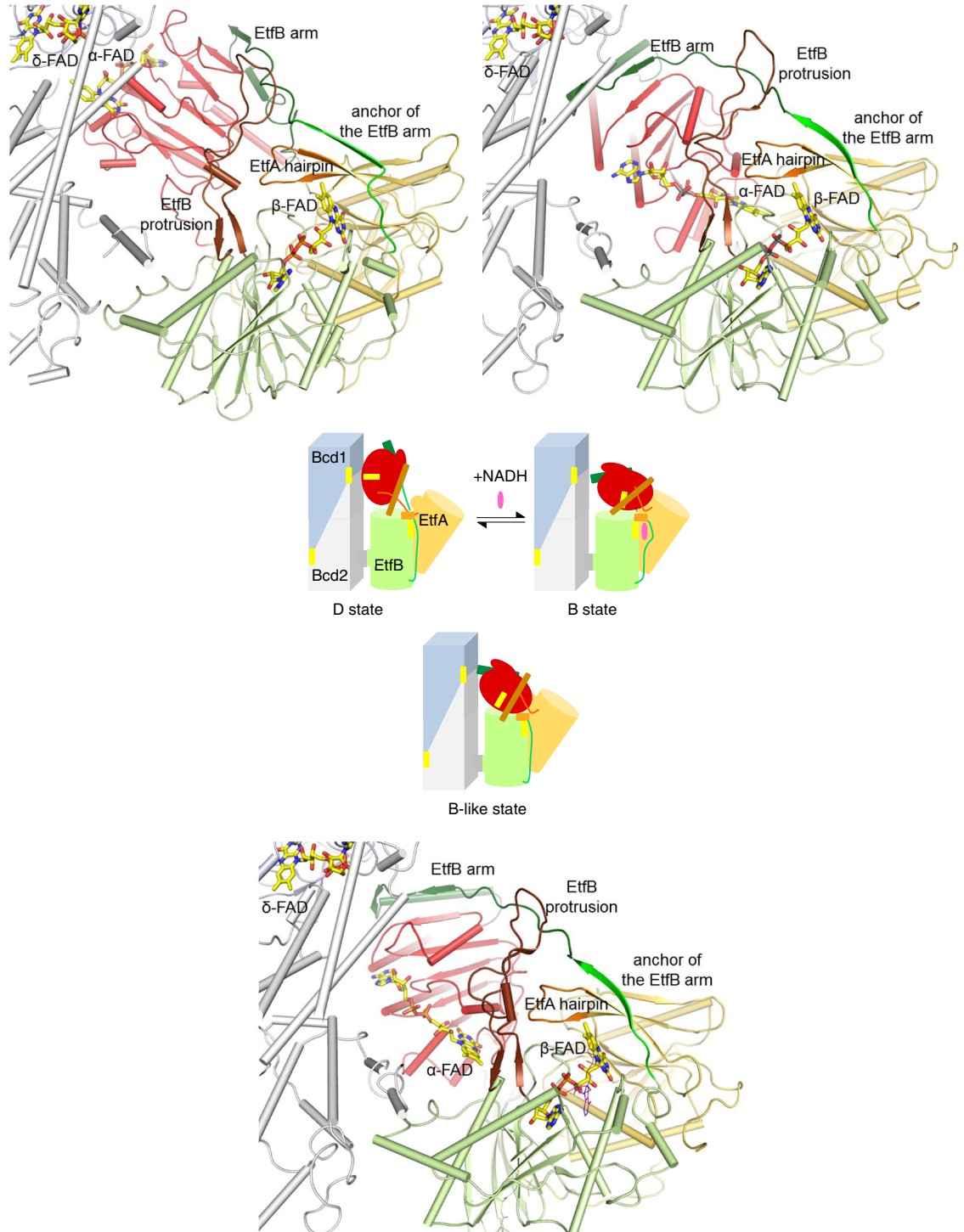

**Fig. 4** Coordination of the rigid-body rotation of domain II. The D, B-like and B states were shown schematically and in structural detail with the focus on domain II (red), the EtfB arm and anchor (green), the EtfB protrusion (brown) and the EtfA hairpin (orange). The postulated signal transmission over 30–40 Å induced by NADH binding or NAD$^+$ release represents the first attempt to understand the coordination of the shuttling process between the B and D states

might be rationalized by an increased linker mobility and a faster rotation of domain II (Fig. 5). In contrast, E165 is also adjacent to the domain I-II linker but its exchange to alanine and aspartate reduces the enzymatic activity. V237 is a residue of the domain III-EtfB arm linker and T18 and P40 are residues of the EtfB protrusion. Their exchange to asparagine, glutamate and leucine decreases the enzymatic activities in a different extent (Fig. 5). In

particular, the P40L exchange might unfavorably influence the conformation and mobility of the EtfB protrusion (see below).

## Discussion

The EtfAB/Bcd structures in the B and D states and the catalytic properties of enzyme variants allowed us to postulate how the

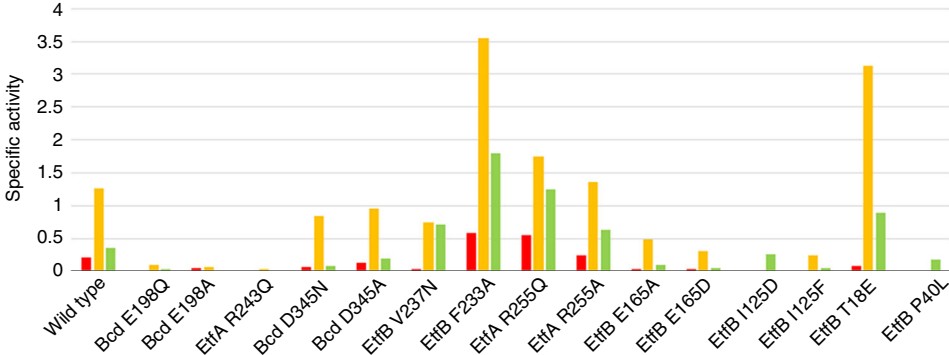

**Fig. 5** Specific activities of the EtfAB/Bcd variants. The specific activities (μmol substrate oxidized $min^{-1} mg^{-1}$ protein) are plotted in red for the NADH bifurcation assay, in orange for the NADH oxidase reaction and in green for the butyryl-CoA dehydrogenase reaction with ferricenium. The enzyme variants can be subdivided into three groups. The first group includes mutations at the Bcd1/Bcd2-domain II interface formed in the D state. Salt bridges are found between Bcd E198−EtfA R243 and Bcd D345−EtfA R243 (see Fig. 3). The second group of mutations addresses the bendable covalent linkers between domain III and the EtfB arm (EtfB F233 and V237) and between domains I and II (EtfA R255). The third group comprises residues that influence the pathway of domain II rotation. EtfB I125 and E165 are located in loop regions contacting domain II. T18 and P40 belong to the EtfB protrusion

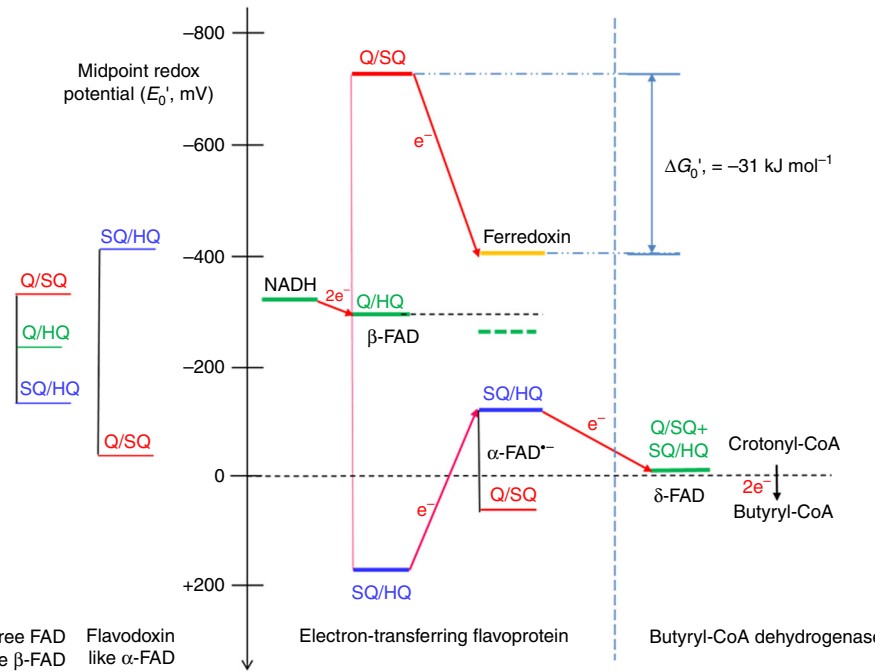

**Fig. 6** Proposed bifurcating electron flow from NADH to crotonyl-CoA and ferredoxin. Q = quinone, SQ = semiquinone, and HQ = hydroquinone forms of FAD. The redox potentials of free FAD[35, 36] and of flavodoxin from *A. fermentans*[37] are shown for comparison. Acceptors of the bifurcating electrons are ferredoxin ($E_0'$ = −405 mV)[26] and the flavodoxin-like α-FAD with a reversed redox potential as that of β-FAD (see text). The green dashed line represents the average of the redox potentials of ferredoxin and α-FAD (SQ/HQ) indicating that the bifurcation NADH + ferredoxin + α-FAD● ⇌ $NAD^+$ + ferredoxin⁻ + α-FADH⁻ is slightly exergonic by ½(−405 −136) − (−279) = +8 mV (0.8 kJ $mol^{-1}$). The α-FADH⁻ swings over to Bcd and reduces δ-FAD with one electron. After repetition of the process, δ-FADH⁻ reduces crotonyl-CoA to butyryl-CoA. The $\Delta G_0'$ value for the one-electron transfer from β-FAD SQ to ferredoxin gives an idea of the energies involved

catalytic events might be coordinated (Fig. 4). Accordingly, the EtfAB/Bcd complex presumably rests in the structurally characterized D state and for initialization FBEB a signal has therefore to be sent over ca. 40 Å from the β-FAD region to the domain II—Bcd interface to induce dissociation. As crotonyl-CoA and ferredoxin (flavodoxin or even $O_2$) bind either spatially far away from β-FAD or only transiently and unspecific, respectively, domain II swinging is presumably triggered by NADH binding or subsequent β-FADH⁻ formation (Supplementary Fig. 3). For hydride transfer from NADH to β-FAD, the nicotinamide moiety has to face the isoalloxazine ring at its *Re*-side which is, however, occupied in the D and B-like states by the segment 222–229

anchoring the EtfB arm. For example, the pyrimidine ring of β-FAD is 3.8 Å apart from EtfB T227 (Supplementary Fig. 2). Thus, productive NADH accommodation will displace the anchor of the EtfB arm by at least 2.0 Å away from the isoalloxazine by which the associated EtfA hairpin (on the top of the nicotinamide) and the EtfB protrusion segments are disengaged (Fig. 4). Simultaneously, the outwards displaced EtfB arm pulls on domain II which becomes dissociated from Bcd and moves from the D to the B state thereby pushing aside the mobile EtfB protrusion. Electron uptake is now feasible. The formation of α-FADH⁻ and β-FADH● perhaps accompanied with the release of $NAD^+$ initiate the switching process from the B to the D state involving

**Table 1 Crystallographic data**

| Crystal | EtfAB/Bcd |
|---|---|
| Crystallization conditions | 0.1 M calcium acetate; 14 % (v/v) PEG 400; 0.1 M MES, pH 6.5 |
| Cryo conditions | 0.1 M calcium acetate; 0.1 M MES, pH 6.5; 30 % PEG 400; 10 μM FAD |
| *Data collection* | |
| Space group | $I4_122$ |
| Wavelength [Å] | 1.0 |
| Resolution range [Å] | 50.0-3.1 (3.2-3.1) |
| Unit cell *a; b; c* [Å] | 177.1, 177.1, 493.2 |
| Redundancy | 7.2 (7.6) |
| Completeness [%] | 98.9 (98.4) |
| $R_{sym}$ [%] | 14.3 (364.1) |
| $I/\sigma(I)$ | 11.1 (0.9) |
| *Refinement* | |
| No. reflections | 69810 (6711) |
| Mol. asym. unit | 2 EtfAB/Bcd |
| No. of atoms: polypeptide, ligands | 14744, 421 |
| $R_{work}$, $R_{free}$ (%) | 20.8 (47.5), 25.9 (51.9) |
| $B_{average}$ (Å²): polypeptide, ligands | 130.3, 150.8 |
| R.m.s. deviation | |
| Bond lengths (Å) | 0.018 |
| Bond angles (°) | 1.87 |
| Ramachandran plot favored, outliers (%) | 92.0, 0.4 |
| PDB ID | 5OL2 |

an inverted sequence of events. Thus, conformational changes of the EtfB protrusion and the EtfB arm unlocks the contact between domain II and the EtfAB base and thereby induces the rotation of the latter into the D state. From there, the electrons flow from α-FADH⁻ to δ-FAD or δ-FADH• (Fig. 2b).

This mechanistic scenario is in line with various experimental data. (1) Domain II can be considered as rigid body connected by two conformationally mobile, solvent-exposed linkers to the EtfAB base (domain I + III). The flavodoxin-like fold of domain II is strongly attached to the EtfB arm such that a pulling force on the latter affects its location (Fig. 4). The superimposed D and B-like states reveal the usage of the mobile linking segments as hinge points in the swinging process. The EtfB arm changes its direction directly after its anchor (at F233) and the EtfB protrusion guides the track of its N-terminal segment perhaps for controlling the rotation of domain II. The covalent domain I-II linker of EtfA is subjected to large rearrangements clearly visible in the electron densities. For example, the distance between K198 and Q203 is 2.5 Å in the D state and 18 Å in the B-like state. The covalent linker in the B state probably has a similar conformation as in the B-like state. (2) Various EtfAB structures show that the EtfB protrusion can adopt highly different conformations which influence the magnitude of the domain II rotation. *C. difficile* and *A. fermentans* EtfB protrusions deviate ca. 5 Å; those between bifurcating *A. fermentans* and non-bifurcating *Methylophilus methylotrophus* EtfAB up to 20 Å[30]. The EtfB protrusion in *M. methylotrophus* EtfAB conformation is oriented towards β-FAD and NADH in the superimposed bifurcating EtfAB complex and would allow a further rotation of domain II towards β-FAD. The distance between the FADs would be ca. 16 Å (compared to 18 Å in the B-like EtfAB conformation). (3) In the structure of the EtfAB-NAD⁺ complex of *A. fermentans*, the ribose-nicotinamide moiety is disordered[24]. Obviously, NAD⁺ but also NADH cannot push aside the anchor of the EtfB arm and occupy its natural binding site parallel to the isoalloxazine. One reason might be that domain II of *A. fermentans* EtfAB is arrested in the B-like

state and not located in the resting D state in the absence of Bcd. (4) Multiple attempts to crystallize EtfAB/Bcd in the presence of NADH failed although crystallization in its absence is well reproducible. This finding argues for a significant conformational rearrangement within the EtfAB/Bcd complex dependent on NADH binding. (5) Site-specific mutagenesis experiments support important functions of the EtfB arm and the EtfB protrusion in the catalytic process (Fig. 5). The catalytically inactive P40L variant appears to modify the conformation of the EtfB protrusion and the F233A variant influences the interactions between the EtfB arm and the EtfAB base which is a very sensitive factor for domain II rotation.

Electron bifurcation requires a cofactor, e.g., a flavin that can be reduced by a two-electron step and oxidized by two one-electron steps or vice versa. This bifurcating cofactor must exhibit 'crossed potentials' i.e. the potential of the semiquinone/ hydroquinone (SQ/HQ) transition must be much higher than that of the quinone/ semiquinone (Q/SQ) transition[12, 34]. This behavior makes sure that after the high-potential ET from HQ to a high-potential acceptor, the remaining semiquinone has a very low potential able to reduce low potential acceptors, such as ferredoxin ($E_0$' = −405 mV)[26] or the SQ of flavodoxin. Interestingly, the one-electron midpoint potentials of free flavin are also crossed; Q/SQ, $E_0$' = −314 mV; SQ/HQ, $E_0$' = −124 mV[35, 36] and their separation ($\Delta E$ = −124 − (−314) = + 200 mV) is increased by interaction with the protein. In flavodoxin, however, the potentials are not crossed; Q/SQ, $E_0$' = −60 mV; SQ/HQ, $E_0$' = −420 mV[37] (Fig. 6).

To establish an energy scheme of the EtfAB/Bcd complex, the two- and one-electron redox potentials of each of the reactants have to be known (Fig. 6). In case of the closely related EtfAB of *Megasphaera elsdenii*, the midpoint potential values at pH 7 have been measured for the flavodoxin-like α-FAD: Q/SQ, $E_0$' = + 81 mV, SQ/HQ, $E_0$' = −136 mV and for β-FAD: Q/HQ, $E_0$' = −279 mV[22]. The two-electron potential of δ-FAD of Bcd is assumed to be close to that of crotonyl-CoA/butyryl-CoA, $E_0$' = −10 mV[25]. The one-electron potentials of β-FAD cannot be determined by the standard redox titration method, because the concentration of the semiquinone SQ of the bifurcating β-FAD is expected to be extremely low; the stability constant $K_s$ = [SQ]² × {[Q] × [HQ]}⁻¹[12] for the quinones in the Rieske/cytochrome b complexes of the Q-cycle was estimated as $10^{−14}$ to $10^{−15}$[38, 39]. In a recent paper, transient absorption spectroscopy (TAS) of the bifurcating FAD of Nfn from *Pyrococcus furiosus* was used to determine the half-life of an artificial generated anionic SQ of FAD as 10 ps ($10^{−11}$ s) from which the one-electron redox potential of FAD, Q/SQ was calculated as −911 mV[21]. This potential is 635 mV lower than the measured two-electron redox potential for FAD: Q/HQ, $E_0$' = −276 mV. Hence the one-electron redox potential of FAD, SQ/HQ amounts to −276 + 635 mV = + 359 mV. Thus the two one-electron redox potentials are separated by $\Delta E$ = 2 × 635 = 1270 mV[21], which at 25 °C (298 K) is converted to log $K_s$ = $\Delta E$ × F × (2.3 RT)⁻¹ = −21.5, $K_s$ = $10^{−21.5}$[12]. For β-FAD of EtfAB, however, such a large separation of the one-electron redox potentials appears not necessary because the low potential acceptor ferredoxin has a 306 mV higher potential than that of Nfn, the proximal [4Fe-4S] cluster ($E_0$' = −711 mV). If we assume a $K_s$ = $10^{−16}$, close to that of ubiquinone, the one-electron redox potentials become separated by $\Delta E \approx$ 900 mV leading for β-FAD, to SQ/HQ, $E_0$' = −279 mV + 450 mV = + 171 mV and to Q/SQ, $E_0$' = −279 mV − 450 mV = −729 mV.

Due to the high redox potential of α-FAD the question arises whether α-FAD (Q/SQ, $E_0$' = + 81 mV) or α-FAD•⁻ (SQ/HQ, $E_0$' = −136 mV) acts as high-potential acceptor. In earlier experiments fully oxidized EtfAB (10 μM) was titrated with increasing concentrations of NADH, showing the reduction of

EtfAB to the stable red anionic semiquinone of α-FAD at 4–6 µM NADH. At 20 µM NADH both α- and β-FAD became reduced to the HQ[24]. This certainly happens also in vivo, where NADH is present. Similarly, the neutral blue semiquinone of flavodoxin (Q/SQ, $E_0' = -60$ mV) is the dominant form in vivo[6]. Hence, most likely α-FAD•⁻ serves as high-potential acceptor for the bifurcation and delivers as α-FADH⁻ one electron to δ-FAD of Bcd, because the redox potential of the SQ/HQ pair is too high.

FBEB starts with a two-electron reduction of β-FAD to β-FADH⁻ by NADH. As soon as the first electron of β-FADH⁻ surmounts the energetic barrier to the semiquinone of α-FAD ($\Delta E = +171 - (-136) = +307$ mV), the extremely low potential semiquinone of β-FAD (β-FADH• or β-FAD•⁻; $E_0' = -729$ mV) reduces ferredoxin ($E_0' = -405$ mV)[26], making the EtfAB partial reaction slightly exergonic as indicated by the green dashed line in Fig. 6, which represents the average of the redox potentials of ferredoxin and α-FAD (SQ/HQ): ½(−405 −136) = −270.5 mV, 8.5 mV more positive than −279 mV, the two-electron redox potential of β-FAD. This endergonic one-electron reduction of α-FAD•⁻ ensures a tight coupling with the exergonic reduction of ferredoxin, as observed experimentally[2]. The formed α-FADH⁻ bound to domain II of EtfAB swings over from the β-FAD (B state) to the δ-FAD (D state) ET mode, which enables a smooth one-electron transfer to δ-FAD, $\Delta E = -10 - (-136) = +126$ mV. Binding of a new molecule of NADH close to β-FAD most likely triggers the return of α-FAD•⁻ to the B-state, where it accepts another electron and swings over to Bcd to generate δ-FADH⁻, which reduces crotonyl-CoA to butyryl-CoA ($E_0' = -10$ mV).

The structures of the D and B states of the EtfAB/Bcd complex show domain II in two distinct orientations by which either a β-FAD to α-FAD or a α-FAD to δ-FAD electron-fire mode is adjusted. Switching between the two states is achieved by rotating α-FADH⁻ (forward) and α-FAD•⁻ (backward) together with the associated domain II over 30 Å which first ensures the shuttling of two reducing equivalents from NADH to crotonyl-CoA. Second, the high redox potential of the α-FAD/α-FAD•⁻ pair of domain II resulting in a α-FAD•⁻ resting state might be also exploited to gate the second electron released by β-FADH• to the weak electron acceptor ferredoxin because α-FADH⁻ cannot accept a further electron. The substantially reduced enzymatic activities of enzyme variants, specifically modified in the two contact regions, substantiate the existence of the two distinct domain II orientations during the catalytic process in solution (Fig. 5). A related "dynamic drive" scenario was established for non-bifurcating EtfAB-dehydrogenase complexes by which the membrane-bound respiratory chain is supplied with reducing equivalents via two single-electron steps from α-FAD•⁻ to quinone. The mobility of domain II carrying α-FAD is applied for sampling a large range of orientations to provide specifically a fast intersubunit ET for structurally distinct dehydrogenase partners[30, 40, 41]. Obviously, the swinging domain II between two redox modules far away from each other is applicable for two different biological purposes.

## Methods

**EtfAB/Bcd production in _Escherichia coli_ and purification**. The EtfAB/Bcd genes from _C. difficile_ (DSM 1296 Type strain; ATCC 9689) are clustered in the genome as _bcd_ (CD 1054), _etfB_ (CD 1055) and _etfA_ (CD 1056). They were cloned in this order into the pASG IBA-3 vector according to manufacturer's instruction (IBA, Göttingen, Germany)[13]. Both applied primers are listed in the Supplementary Table 2. The proteins with a C-terminal Strep-Tag II at EtfA were subsequently overproduced together in _E. coli_ BL21(DE3), (New England Biolabs, Ipswich, MA, USA) using anhydrotetracycline (200 µg l⁻¹, IBA). Cells were grown in 4 L culture of Luria Bertani broth containing ampicillin (100 µg ml⁻¹). EtfAB/Bcd production was induced at an optical density (OD$_{600nm}$) of 0.5. After growing overnight at room temperature, cells were harvested (15 g wet mass), suspended in 50 mM Tris/HCl buffer containing 150 mM NaCl, pH 7.5 (buffer A) and disrupted by a passage through a French press at 120 MPa. Cell lysate was centrifuged at 30,000 × g for 1 h

at 4° C. The supernatant was loaded onto a 10 ml Strep-Tag II (IBA) column and was washed with 150 ml buffer A. The recombinant protein was eluted with 15 ml of 2.5 mM D-desthiobiotin in buffer A and concentrated using a Centricon filter tube (50 kDa cutoff, EMD Millipore). After incubation with an excess of FAD at 4° C overnight, protein and unbound FAD was separated using a Superdex 200 pg column (GE Healthcare); only peak fractions were used for crystallization. The protein homogeneity was checked by SDS-PAGE after purification.

**Generation of site-specific EtfAB/Bcd variants**. Multiple variants of the EtfAB/Bcd complex were generated by site-directed mutagenesis using overlapping primers (see Supplementary Table 2) and the polymerase chain reaction (PCR) method. Subsequently, plasmid pASG IBA-3 containing the gene of an EtfAB/Bcd variant was DpnI digested overnight, the digested plasmid purified with a PCR purification kit (Thermo Scientific), transformed into _E coli_ DH5α (Invitrogen, Darmstadt, Germany) and plated on to LB agar containing ampicillin (100 µg ml⁻¹). The plasmid was isolated and sequenced for verifying the correctness of the mutation. Finally, the corresponding plasmid was transformed into _E. coli_ BL21(DE3), and the EtfAB/Bcd variant produced and purified as described for the wildtype protein. The EtfA R136Q/K/M variant showed little or no overproduction, perhaps due to instability.

**Enzyme assays**. The anaerobic bifurcation activity of EtfAB/Bcd was measured in a 500 µl cuvette ($d = 1$ cm) containing 50 mM potassium phosphate, pH 6.8, 250 µM NADH, 100 µM crotonyl-CoA, 1.5 µM EtfAB/Bcd, 5 µM ferredoxin from _Clostridium tetanomorphum_[24] and crude [FeFe] hydrogenase (30 µg ml⁻¹) from _Clostridium pasteurianum_[24]. The decrease in NADH concentration was monitored at 340 nm, $\varepsilon = 6.3$ mM⁻¹ cm⁻¹[42]. The Bcd activity was measured under aerobic conditions using 0.35 µM EtfAB/Bcd, 0.2 mM ferricenium hexafluorophosphate (Fc⁺) and 50 µM butyryl-CoA in 50 mM potassium phosphate, pH 6.8. The decrease of absorbance was followed at 310 nm, $\varepsilon = 2 \times 4.3$ mM⁻¹ cm⁻¹, because 2 mol of Fc⁺ are required to oxidize 1 mol of butyryl-CoA[43]. The NADH oxidase activity of EtfAB/Bcd was measured in the presence of oxygen and in the absence of crotonyl-CoA using 0.2 µM EtfAB/Bcd and 150 µM NADH in 50 mM potassium phosphate, pH 6.8. Prior to the reaction, the phosphate buffer was bubbled with oxygen gas. The assays were repeated with the EtfAB/Bcd variants. The data were plotted in bar graphs using Graphpad Prism 5.

**X-ray structure analysis of the EtfAB/Bcd complex**. EtfAB/Bcd was concentrated to 20 mg ml⁻¹ in 10 mM MOPS and 1 mM FAD, pH 7.0. Crystallization experiments were performed with the sitting drop method at 4 and 18 °C using a CrystalMation™ system from Rigaku and commercially available screens. Initial conditions were found in the JBScreen PACT + + HTS which were further optimized (see Table 1). Promising-looking crystals of the EtfAB/Bcd complex of _C. difficile_ grew rather reproducible but only about 1% diffracted to a resolution of <3.5 Å. The diffraction power normally varies between 5 and 8 Å resolution. After extended screening, X-ray diffraction data were collected at the PXII beamline of the Swiss-Light-Source in Villigen with a crystal diffracting to 3.1 Å resolution and processed with XDS[44]. Phases were determined by the molecular replacement method[45] using EtfAB and Bcd of _A. fermentans_ separately as model. Model errors were corrected within COOT[46]. The refinement was performed with REFMAC5[47], PHENIX[48] and BUSTER (Phaser, Global Phasing Ltd.). The final $R/R_{free}$ values were 20.8/25.9 %. Data quality was checked by MOLPROBITY[49] and refinement statistics were listed in Table 1. Figures 2–4 and Supplementary Figs. 1–4 were generated with PYMOL (Schrödinger, LLC).

**Data availability**. Atomic coordinates and structure factors for the crystal structure of the EtfAB/Bcd domplex are deposited in the protein databank under the accession code 5OL2. Other data are available from the corresponding authors upon reasonable request.

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

## Acknowledgements

We thank Rolf K. Thauer (MPI of terrestrial Microbiology) for support and helpful discussions, Hartmut Michel (MPI of Biophysics) for continuous support, Yvonne Thielmann and Barbara Rathmann of the Core Center (MPI of Biophysics) for performing crystallization screenings and the staff of the Swiss-Light-Source, Villigen, for help during data collection, Seigo Shima (MPI of terrestrial Microbiology) for help in flavin identification. This work was funded by the Max-Planck Society, the German Research Foundation (DFG) and SYNMIKRO of the Philipps-Universität Marburg.

## Author contributions

W.B. and U.E. directed the research and designed the study. T.S. provided the clone, N.P. C. expressed the genes, purified the enzyme and made the mutations. J.K.D. crystallized the enzyme and determined the structure. All authors interpreted the data and wrote the manuscript.

## Additional information

**Competing interests:** The authors declare no competing financial interests.

