## [Peer Review File · Nature Communications]

Reviewers' comments:

Reviewer #1 (Remarks to the Author):

Electron bifurcation is a fundamental catalytic strategy to couple energetic uphill and downhill reactions. The authors of this study have made seminal contributions to understand the mechanism and physiological relevance of a novel flavin based type of electron bifurcation. Here, they solved the X-ray structure of the EtfAB/Bcd complex from *Clostridium difficile*. Comparison with the corresponding structure of *Acidaminococcus fermentans* revealed a considerable difference in the conformation of domain II of subunit EtfA. FAD- $\alpha\alpha$ in domain II was close to FAD- $\beta\beta$ in the *C. difficile* structure while it was close to FAD- $\delta\delta$ in the *A. fermentans* structure. The authors propose a rigid body movement of domain II of EtfA as the structural basis for coordination of electron transfer events and thus of electron bifurcation in this enzyme. The model is validated by structure guided site directed mutagenesis. The conformational switch between D and B state is reminiscent of the movement of the Rieske FeS domain in respiratory complex III. This study is a major step forward to unravel a fundamental principle in biochemistry and is well suited for the broadly interested readership of nature communications.

Minor remarks:

Please check residue numbering Arg253/243 in Fig.3/text.

Would it be possible to show (proposed) NADH and Ferredoxin binding sites in Fig.2?

Please discuss more clearly the mobility of domain II in non-bifurcating EtfAB dehydrogenases in relation to your proposed model. Since domain mobility is highlighted as the structural basis for bifurcation (enabling electron transfer and gating) it might confuse the reader to end the discussion with a reference to domain mobility that permits optimal orientation for electron transfer in an energetically simpler setting without bifurcation.

Reviewer #2 (Remarks to the Author):

The ms by Demmer et al. describes the structure of the EtfAB/Bcd complex from *Clostridium difficile* as well as activity essays of enzymatic partial reactions in a number of site-directed mutants of the enzyme. The obtained structure is compared to a previous structure obtained for the individual enzymes EtfAB and Bcd from *A. fermentans* yielding the result that domain II of subunit A can swing back and forth between two extreme configurations denoted as D- and B-states. The ensemble of the presented data constitute a major step ahead in our understanding of flavin-based electron bifurcation, an only recently discovered energy-conserving mechanism, the biological importance of which can hardly be overstated. I therefore consider this ms as perfectly suited for publication in Nature Communications and I strongly recommend its acceptance in this journal.

I have only 3 minor comments pertaining to the discussion section:

- lines 306-308: I don't think that it is correct to say that the reduction of ferredoxin " stabilizes the semiquinone on alpha-FAD". Given the electron distances you estimate, back and forth electron transport between both beta-FADred and alpha-FADox and beta-FADsemi and ferredoxin will be ultrarapid and only the fraction of time each of the two electrons spends on its respective donor/acceptor will depend on the mutual redox potentials. What stabilizes the total reaction in my view is the fact that the average of the potentials of ferredoxin and alpha-FADox/semi (-162 mV) is more positive than the -279 mV of the 2-electron transition of beta-FAD.

- lines 312-314 (and before, where the interaction surfaces are discussed): You say that the redox potential of alpha-FAD could decrease by about 100 mV due to interaction ... with hydrophobic residues of Bcd. I am worried that this is a highly anti-thermodynamic affirmation. If the swing would take the alpha-FAD to lower potentials, i.e. to higher energy levels, it just wouldn't swing

there since it wants to be in the lowest energy state (as we all do, don't we?).

As I have discussed with one of the authors previously, I actually don't think that you need the lower redox potential for your model to work if you consider that it is the alpha-FADsemi/red transition rather than the alpha-FADox/semi one which is pertinent in-vivo. A redox centre with a potential of +81 mV is almost certainly always reduced in the cytoplasm. I would therefore consider it highly likely that the first electron from the bifurcating beta-FAD reduces alpha-FADsemi to alpha-FADred. The onward reaction to delta-FAD is then automatically downhill. All that changes with respect to your scheme is that the uphill reaction for the first electron will be even more uphill and that the driving force for the total 2-electron transfer from beta-FAD to ferredoxin and alpha-FAD will be less exergonic but still exergonic (50 mV downhill in my scheme). This is almost perfectly the value you see in the Rieske/b complexes and in Nfn (if one uses the values given in the Lubner-paper). But that is just my opinion and the authors can disregard it. I am just uncomfortable with the notion that the domain would by itself swing to a position with a higher energy. Smells like Perpetuum Mobile to me.

- line 321: "... bifurcation ... proceeds within ps". How do you know that? If you take that information from the Lubner paper, I would hold that what they look at is a completely different reaction. The ps-reaction seen in this paper is forward electron transfer from a UV-flash induced reduced state of the flavin, that is, an artificial electron transfer reaction. Bifurcation is less about electron transfer rates (determined by distances and redox potential differences according to the Moser-model) than about redox-equilibration of two distinct but coupled electron transfers. The literature in photosynthesis research is full of examples where redox-equilibration is found 100-1000 times slower than direct electron transfer rates. I would not mention this ps-value here since you have no evidence for that.

Given the details I discussed above, I guess I basically don't need to sign this report since my identity is likely "transparent" to the authors.

Anyway

Wolfgang Nitschke

Reviewer #3 (Remarks to the Author):

This manuscript describes the structure of the hetero-decametric complex (ETFAB/Bcd)₄ formed by electron transfer flavoprotein (ETF) and butyryl-CoA dehydrogenase (Bcd) from the obligate anaerobic bacterium *Clostridium difficile*. This structure is potentially of broad interest as it is the first structure of an ETF capable of electron bifurcation in complex with its cognate primary flavoprotein dehydrogenase. Superposition onto this structure of the separate structures (e.g. solved outside a complex) of the bifurcating ETFAB from *Acidominococcus fermentans* suggest that domain II of ETFAB can adopt two conformations. The first (B-conformation) brings the -FAD prosthetic group of ETFAB close enough to the -FAD, which serves the immediate electron acceptor for the co-substrate NADH, to allow productive electron of a single high potential electron. The second low potential electron is transferred to a ferredoxin in this electron bifurcation reaction. In the alternate (D-conformation) the -FAD semiquinone is brought into electron transfer range of the -FAD of Bcd which serves as electron donor for the co-substrate crotonyl-CoA. This structural information is integrated with new information about the reduction potentials of the -FAD semiquinones, derived from transient absorption spectroscopy of a related system, to generate a refined model of electron bifurcation. The authors then go on to test the proposed mechanism by making a series of site specific variants of both ETFAB and Bcd which they probe with three different enzyme assays.

Although the role of domain movement in gating electron bifurcation is well known from extensive structural and biophysical studies of bc1 complexes the phenomenon has only recently been

described in anaerobic ETFs and should be of interest to general audience. However I wonder if the work might have greater impact if the authors gave some further physiological context to emphasise how this process underpins the anaerobic lifestyle and any potential biotechnological applications.

Despite the modest resolution (3.1Å) of the complex the structural study and associated molecular modelling is convincing and contributes to a plausible model although the Figure 4 is quite hard to relate to the mutagenesis study (see detailed comments below). As presented the mutagenesis data does not add as much value as it could for two reasons. Firstly, it is not clearly explained what the different assays are measuring and what might affect activity. This in turn leads to a very narrow interpretation of the result that focuses almost exclusively on the reasons for enhanced activity in the ETFAB R255 and F233 mutants. Why for instance do mutants that destabilise the ETFAB/Bcd interface decrease NADH oxidase activity – might it be that molecular oxygen can only react with the -FAD (and not -FAD) hydroquinone in the absence of crotonyl-CoA? And is it that ferricenium cannot access the -FAD to act as an electron acceptor in the reverse reaction when the complex is locked in the D-conformation? This kind of details may not be entirely obvious to general readership and need to be made more explicit if this work is to be accessible to a wider audience and published in Nature Communications.

Some specific issues that require attention:-

Line 205: Should be non-bifurcating human ETFAB/MCAD complex

Line 206: As expected the activity of the resulting variants measured in three ways was drastically reduced – please see general comments above.

Line 215 V236 – is V237 in Figure 6?

Line 304: should be +304 mV – electron is going in other direction

Line 313: The decreased stability and lower redox potential on the -FAD semiquinone in the D state relative to the B-state is only attributed to the lower dielectric constant that results from shielding from eBcd hydrophobic residues. Is there also a contribution from the binding energy of the Glu201 and Glu198 interactions (see comments on Figure 6)

Figure 3 Arg253 is mislabelled it should be Arg243 (cf text and Figure 5)

Figure 4 Could the authors find a way to locate and identify the residues that are mutated and reported in Figure 5

Figure 5 Could the authors reorder the mutants on the X-axis to make it easier for the reader: Group 1 (Bcd E198 and D345 and ETFA R243) Group 2 (Linker domain) Group 3 (Protrusion) Group 4 (Anchor). Some text under each group might help the reader make the link to Figure 4

Figure 6 It might be instructive to annotate this figure with some indicative Gibbs free energies

Reviewer #1 (Remarks to the Author):

Electron bifurcation is a fundamental catalytic strategy to couple energetic uphill and downhill reactions. The authors of this study have made seminal contributions to understand the mechanism and physiological relevance of a novel flavin based type of electron bifurcation. Here, they solved the X-ray structure of the EtfAB/Bcd complex from Clostridium difficile. Comparison with the corresponding structure of Acidaminococcus fermentans revealed a considerable difference in the conformation of domain II of subunit EtfA. FAD- α in domain II was close to FAD- β in the C. difficile structure while it was close to FAD- δ in the A. fermentans structure. The authors propose a rigid body movement of domain II of EtfA as the structural basis for coordination of electron transfer events and thus of electron bifurcation in this enzyme. The model is validated by structure guided site directed mutagenesis. The conformational switch between D and B state is reminiscent of the movement of the Rieske FeS domain in respiratory complex III. This study is a major step forward to unravel a fundamental principle in biochemistry and is well suited for the broadly interested readership of nature communications.

Minor remarks:

Please check residue numbering Arg253/243 in Fig.3/text.

We changed in Fig.2 Arg253 to Arg243.

Would it be possible to show (proposed) NADH and Ferredoxin binding sites in Fig.2?

We indicated the NADH and ferredoxin binding site in Fig. 2 and included a supplementary Fig. 2 that show a more detail model.

Please discuss more clearly the mobility of domain II in non-bifurcating EtfAB dehydrogenases in relation to your proposed model. Since domain mobility is highlighted as the structural basis for bifurcation (enabling electron transfer and gating) it might confuse the reader to end the discussion with a reference to domain mobility that permits optimal orientation for electron transfer in an energetically simpler setting without bifurcation.

We changed the end of the discussion to clarify that a related domain rearrangement is used for different biological purposes.

Reviewer #2 (Remarks to the Author):

The ms by Demmer et al. describes the structure of the EtfAB/Bcd complex from Clostridium difficile as well as activity essays of enzymatic partial reactions in a number of site-directed mutants of the enzyme. The obtained structure is compared to a previous structure obtained for the individual enzymes EtfAB and Bcd from A. fermentans yielding the result that domain II of subunit A can swing back and forth between two extreme configurations denoted as D- and B-states. The ensemble of the presented data constitute a major step ahead in our understanding of flavin-based electron bifurcation, an only recently discovered energy-conserving mechanism, the biological importance of which can hardly be overstated. I therefore consider this ms as perfectly suited for publication in Nature Communications and I strongly recommend its acceptance in this journal.

I have only 3 minor comments pertaining to the discussion section:

- lines 306-308: I don't think that it is correct to say that the reduction of ferredoxin " stabilizes the semiquinone on alpha-FAD". Given the electron distances you estimate, back and forth electron transport between both beta-FADred and alpha-FADox and beta-FADsemi and ferredoxin will be ultrarapid and only the fraction of time each of the two electrons spends on its respective donor/acceptor will depend on the mutual redox potentials. What stabilizes the total reaction in my view is the fact that the average of the potentials of ferredoxin and alpha-FADox/semi (-162 mV) is more positive than the -279 mV of the 2-electron transition of beta-FAD.

The discussion about the mechanism was substantially changed which follows the suggestions of the reviewer and also takes into account this specific point. We removed the section "stabilizes the semiquinone on alpha-FAD" . We also integrated the idea of the reviewer that the total reaction is stabilized that the potentials of ferredoxin and α -FADsemi/red (-270.5 mV) is more positive than the 2-electron reduction of β -FAD by NADH (-279 mV).

Line 329-335

As soon as the first electron of β -FADH⁻ surmounts the energetic barrier to the semiquinone of α -FAD ($\Delta E = +171 - (-136) = +307$ mV), the extremely low potential semiquinone of β -FAD (β -FADH[•] or β -FAD^{•-}; $E_0' = -729$ mV) reduces ferredoxin ($E_0' = -405$ mV²⁶), making the EtfAB partial reaction slightly exergonic as indicated by the green dashed line in Fig. 6, which represents the average of the redox potentials

of ferredoxin and α -FAD (SQ/HQ): $\frac{1}{2}(-405 - 136) = -270.5$ mV, 8.5 mV more positive than -279 mV, the two-electron redox potential of β -FAD.

- lines 312-314 (and before, where the interaction surfaces are discussed): You say that the redox potential of alpha-FAD could decrease by about 100 mV due to interaction ... with hydrophobic residues of Bcd. I am worried that this is a highly anti-thermodynamic affirmation. If the swing would take the alpha-FAD to lower potentials, i.e. to higher energy levels, it just wouldn't swing there since it wants to be in the lowest energy state (as we all do, don't we?). But that is just my opinion and the authors can disregard it. I am just uncomfortable with the notion that the domain would by itself swing to a position with a higher energy. Smells like Perpetuum Mobile to me.

The energy for decreasing the redox potential of α -FAD \bullet^- or α -FADH $^-$ is provided by the binding energy between domain II and the Bcd1-Bcd2 interface. Therefore we think that a decrease of the redox potential is, in principle, possible and will not violate the law of energy conservation.

As I have discussed with one of the authors previously, I actually don't think that you need the lower redox potential for your model to work if you consider that it is the alpha-FADsemi/red transition rather than the alpha-FADox/semi one which is pertinent in-vivo. A redox centre with a potential of +81 mV is almost certainly always reduced in the cytoplasm. I would therefore consider it highly likely that the first electron from the bifurcating beta-FAD reduces alpha-FADsemi to alpha-FADred. The onward reaction to delta-FAD is then automatically downhill. All that changes with respect to your scheme is that the uphill reaction for the first electron will be even more uphill and that the driving force for the total 2-electron transfer from beta-FAD to ferredoxin and alpha-FAD will be less exergonic but still exergonic (50 mV downhill in my scheme). This is almost perfectly the value you see in the Rieske/b complexes and in Nfn (if one uses the values given in the Lubner-paper).

We have accepted the proposal of the reviewer (although it is not finally proved) to consider the semi state of α -FAD as the resting state and the state in the cell.

Therefore, the new scheme takes this reinterpretation into account.

Lines 318-327:

Due to the high redox potential of α -FAD the question arises whether α -FAD (Q/SQ, $E_0' = +81$ mV) or α -FAD \bullet^- (SQ/HQ, $E_0' = -136$ mV) acts as high potential acceptor. In earlier experiments fully oxidized EtfAB (10 μ M) was titrated with increasing concentrations of NADH, showing the reduction of EtfAB to the stable red anionic semiquinone of α -FAD at 4-6 μ M NADH. At 20 μ M NADH both α - and β -FAD became reduced to the HQ²⁴. This certainly happens also *in vivo*, where NADH

is present. Similarly, the neutral blue semiquinone of flavodoxin (Q/SQ, $E_0' = -60$ mV) is the dominant form *in vivo*⁶ (W. Buckel, unpublished). Hence, most likely α -FAD \cdot^- serves as high potential acceptor for the bifurcation and delivers as α -FADH \cdot^- one electron to δ -FAD of Bcd, because the redox potential of the second electron is too high.

- line 321: "... bifurcation ... proceeds within ps". How do you know that? If you take that information from the Lubner paper, I would hold that what they look at is a completely different reaction. The ps-reaction seen in this paper is forward electron transfer from a UV-flash induced reduced state of the flavin, that is, an artificial electron transfer reaction. Bifurcation is less about electron transfer rates (determined by distances and redox potential differences according to the Moser-model) than about redox-equilibration of two distinct but coupled electron transfers. The literature in photosynthesis research is full of examples where redox-equilibration is found 100-1000 times slower than direct electron transfer rates. I would not mention this ps-value here since you have no evidence for that.

We have changed this section according to the advice of the reviewer.

The stability constant $K_s = [\text{SQ}]^2 \times \{[\text{Q}] \times [\text{HQ}]\}^{-1}$ for the quinones in the Rieske/cytochrome b complexes of the Q-cycle was estimated as 10^{-14} to 10^{-15} . For the FAD + FADH $_2$ to 2FADH \cdot equilibrium a $K_s = 10^{-21.5}$ (10 ps) was obtained for the Nfn of *Pyrococcus furiosus*. This value appears to be too low for a redox equilibration process. For our scheme we used a K_s of 10^{-16} which is not experimentally proven but plausibly reflect the present incomplete knowledge. A change of K_s by factor of 100 would not dramatically affect the scheme.

Lines 300-317

The one-electron potentials of β -FAD cannot be determined by the standard redox titration method, because the concentration of the semiquinone of the bifurcating β -FAD is expected to be extremely low; with a stability constant $K_s = [\text{SQ}]^2 \times \{[\text{Q}] \times [\text{HQ}]\}^{-1}$.¹² K_s for the quinones in the Rieske/cytochrome b complexes of the Q-cycle was estimated as 10^{-14} to 10^{-15} .^{38,39} In a recent paper, transient absorption spectroscopy (TAS) of the bifurcating FAD of Nfn from *Pyrococcus furiosus* was used to determine the half-life of an artificial generated anionic SQ of FAD as 10 ps (10^{-11} s) from which the one-electron redox potential of FAD, Q/SQ was calculated as

-911 mV^{21} . This potential is 635 mV lower than the measured two-electron redox potential for FAD; Q/HQ, $E_0' = -276 \text{ mV}$. Hence the one-electron redox potential of FAD, SQ/HQ amounts to $-276 + 635 \text{ mV} = +359 \text{ mV}$. Thus the two one-electron redox potentials are separated by $\Delta E = 2 \times 635 = 1270 \text{ mV}^{21}$, which at 25 °C (298 K) is converted to $\log K_s = \Delta E \times F \times (2.3 \text{ RT})^{-1} = -21.5$, $K_s = 10^{-21.5}$.¹² For β -FAD of EtfAB, however, such a large separation of the one-electron redox potentials appears not necessary because the low potential acceptor ferredoxin has a 306 mV higher potential than that of Nfn, the proximal [4Fe-4S] cluster ($E_0' = -711 \text{ mV}$). If we assume a $K_s = 10^{-16}$, close to that of ubiquinone, the one-electron redox potentials become separated by $\Delta E \approx 900 \text{ mV}$ leading for β -FAD, to SQ/HQ, $E_0' = -279 \text{ mV} + 450 \text{ mV} = +171 \text{ mV}$ and to Q/SQ, $E_0' = -279 \text{ mV} - 450 \text{ mV} = -729 \text{ mV}$.

Reviewer #3 (Remarks to the Author):

This manuscript describes the structure of the hetero-decametric complex (ETFAB/Bcd)₄ formed by electron transfer flavoprotein (ETF) and butyryl-CoA dehydrogenase (Bcd) from the obligate anaerobic bacterium Clostridium difficile. This structure is potentially of broad interest as it is the first structure of an ETF capable of electron bifurcation in complex with its cognate primary flavoprotein dehydrogenase. Superposition onto this structure of the separate structures (e.g. solved outside a complex) of the bifurcating ETFAB from Acidaminococcus fermentans suggest that domain II of ETFAB can adopt two conformations. The first (B-conformation) brings the -FAD prosthetic group of ETFAB close enough to the -FAD, which serves the immediate electron acceptor for the co-substrate NADH, to allow productive electron of a single high potential electron. The second low potential electron is transferred to a ferredoxin in this electron bifurcation reaction. In the alternate (D-conformation) the -FAD semiquinone is brought into electron transfer range of the -FAD of Bcd which serves as electron donor for the co-substrate crotonyl-CoA. This structural information is integrated with new information about the reduction potentials of the -FAD semiquinones, derived from transient absorption spectroscopy of a related system, to generate a refined model of electron bifurcation. The authors then go on to test the proposed mechanism by making a series of site specific variants of both ETFAB and Bcd which they probe with three different enzyme assays.

Although the role of domain movement in gating electron bifurcation is well known from extensive structural and biophysical studies of bc1 complexes the phenomenon has only recently been described in anaerobic ETFs and should be of interest to general audience.

However I wonder if the work might have greater impact if the authors gave some further physiological context to emphasise how this process underpins the anaerobic lifestyle and any potential biotechnological applications.

We have added a paragraph to the introduction, in which the importance of flavin-based electron bifurcation for energy conservation in *Clostridium difficile* and in clostridia, in general, is explained. In addition, it is mentioned that the biotechnological production of short chain fatty acids mainly relies on electron bifurcation.

Clostridia thrive from the fermentation of organic compounds to acetate, short chain fatty acids, CO₂, ammonia, and H₂. Thereby carbohydrates, alcohols and amino acids are oxidized to “energy rich” CoA thioesters, which phosphorylate ADP to ATP, a process called substrate level phosphorylation (SLP). The oxidants or electron acceptors are acetate and amino acids which are reduced to butyrate and carboxylic acids with the carbon skeleton of the parent amino acid., *Clostridium difficile*, recently reclassified as *Clostridioides difficile* [Lawson et al. 2016], is able to ferment 2 serine via pyruvate to 2 CO₂ and butyrate, whereas leucine is oxidized to isovalerate and CO₂ concomitant with the reduction of 2 leucine to 2 isocaproate [Kim et al. 2006, Neumann-Schaal et al. 2015]. In each pathway only one ATP can be conserved by SLP via butyryl-CoA or isovaleryl-CoA, respectively, whereas thermodynamic calculations allow twice as much. Indeed, contrary to earlier views, strict anaerobic bacteria are able to perform electron transport phosphorylation (ETP) in which ferredoxin plays the central role. Reduced ferredoxin can be regarded also as an “energy rich” compound, which in many fermenting anaerobes forms H₂ and synthesizes ATP via a Na⁺-pumping ferredoxin-NAD⁺ reductase (Rnf) and a Na⁺-dependent F₁F₀ ATPase^{5,6}. Ferredoxin is reduced by oxidative decarboxylation of 2-oxo acids and the recently discovered flavin-based electron bifurcation (FBEB) [3]. Thus SLP and ETP with Rnf and FBEB make the anaerobes as efficient energy converters as mitochondria. *Clostridium kluyveri* synthesizes butyrate (C4), caproate (C6), caprylate (C8) and H₂ from ethanol and acetate, which is called chain elongation and applied to convert waste from alcoholic fermentations into valuable fatty acids. In this process FBEB is responsible for hydrogen production and for 60% of the conserved energy, which underlines its biotechnological importance [Angenent et al. 2016].

- Lawson, P. A., Citron, D. M., Tyrrell, K. L., and Finegold, S. M. (2016) Reclassification of *Clostridium difficile* as *Clostridioides difficile* (Hall and O'Toole 1935) Prevot 1938, *Anaerobe* 40, 95-99.
- Neumann-Schaal, M., Hofmann, J. D., Will, S. E., and Schomburg, D. (2015) Time-resolved amino acid uptake of *Clostridium difficile* 630 Δ erm and concomitant fermentation product and toxin formation, *BMC Microbiol* 15, 281.
- Angenent L. T., Richter H., Buckel W., Spirito C. M., Steinbusch K. J. J., Plugge C. M., Strik D. P. B. T. B., Grootsholten T. I. M., Buisman C. J. N. and Hamelers H. V. M. (2016). Chain elongation with reactor microbiomes: open-culture biotechnology to produce biochemicals. *Environmental Science & Technology*, Vol. 50, No. 6, pp. 2796-2810.

Despite the modest resolution (3.1Å) of the complex the structural study and associated molecular modelling is convincing and contributes to a plausible model although the Figure 4 is quite hard to relate to the mutagenesis study (see detailed comments below).

We subdivided the enzyme variants into three groups and explained them in the legend of Fig. 5 (see also supplementary Figure 3 in which the location of the mutations is marked).

*As presented the mutagenesis data does not add as much value as it could for two reasons. Firstly, it is not clearly explained what the different assays are measuring and what might affect activity. This in turn leads to a very narrow interpretation of the result that focuses almost exclusively on the reasons for enhanced activity in the ETFAB R255 and F233 mutants. **This kind of details may not be entirely obvious to general readership and need to be made more explicit if this work is to be accessible to a wider audience and published in Nature Communications.***

The assays were described in the Methods section under “Enzyme assays”. We have presented and interpreted all mutations in section “Site-specific mutations in the EtfAB/Bcd complex”. Of course, we agree with the reviewer that for several enzyme variants the kinetic measurements involving apparently one active site only cannot be straightforward interpreted. The presented work, however, indicated that the conformational position of domain II modifies NADH oxidase and butyryl-CoA dehydrogenase activity. Even so, the large difference found between different enzyme variants is a surprise. We already considered this point in the legend of supplementary Table 1 as indirectly suggested by the reviewer (see above in bold).

Why for instance do mutants that destabilise the ETFAB/Bcd interface decrease NADH oxidase activity – might it be that molecular oxygen can only react with the -FAD (and not -FAD) hydroquinone in the absence of crotonyl-CoA?

We found out that the conformation of the anchor of the EtfB arm is influenced from the location/mobility of domain II which, in turn, depends on the mentioned mutations. Simultaneously, the anchor conformation also determines the affinity for binding NADH and its oxidation kinetics. We do not know how the mutations change the O₂ access.

And is it that ferricenium cannot access the -FAD to act as an electron acceptor in the reverse reaction when the complex is locked in the D-conformation?

We do not have any structure of an enzyme variant. Therefore, information about the access is difficult to provide. Moreover, we do not know whether the reduced δ -FAD donates one or two electron to α -FAD (or even further) and ferricenium uptakes the electron from there.

Some specific issues that require attention:-

Line 205: Should be non-bifurcating human ETFAB/MCAD complex

corrected according to the proposal

Line 206: As expected the activity of the resulting variants measured in three ways was drastically reduced – please see general comments above.

We added in the supplementary data a paragraph which comments the kinetic measurements. We agree with the concerns of the reviewer. Almost all mutations have a significant effect and some of them can be rationalized rather confidently. But some activities of the NADH oxidase and butyryl-CoA dehydrogenase partial results are elusive. Perhaps much more mutations are necessary to end up with a congruent picture. Currently, only vague speculations are possible.

Line 215 V236 – is V237 in Figure 6?

We changed V236 to Val237.

Line 304: should be +304 mV – electron is going in other direction

The reviewer is right, but this part was substantially rewritten according to the proposals of reviewer 2.

Line 313: The decreased stability and lower redox potential on the -FAD semiquinone in the D state relative to the B-state is only attributed to the lower dielectric constant that results from shielding from eBcd hydrophobic residues. Is there also a contribution from the binding energy of the Glu201 and Glu198 interactions (see comments on Figure 6)

The interactions between Bcd1 Glu210 and EtfA Arg217 as well as between Bcd1 Glu198 and EtfA Arg243 are presumably important for the Bcd – domain II binding affinity. Glu210 is too far away from the isoalloxazine ring and should not influence the redox potential of α -FAD in contrast to Glu198 which is adjacent to the isoalloxazine ring. Nevertheless, the hydrophobic residues are clearly dominant.

Figure 3 Arg253 is mislabelled it should be Arg243 (cf text and Figure 5)

We have corrected this error.

Figure 4 Could the authors find a way to locate and identify the residues that are mutated and reported in Figure 5

We marked the mutated residues in a new supplementary Fig. 3.

Figure 5 Could the authors reorder the mutants on the X-axis to make it easier for the reader: Group 1 (Bcd E198 and D345 and EtfA R243) Group 2 (Linker domain) Group 3 (Protrusion) Group 4 (Anchor). Some text under each group might help the reader make the link to Figure 4

We subdivided the mutations into three groups and explained them in the legend of Fig. 5 (see also supplementary Figure 3).

Figure 6 It might be instructive to annotate this figure with some indicative Gibbs free energy

We have added the Gibbs free energy for the ET from β -FADH• to ferredoxin as an example.

REVIEWERS' COMMENTS:

Reviewer #1 (Remarks to the Author):

My concerns have been adequately addressed.

Reviewer #2 (Remarks to the Author):

The ms by Demmer et al. has been substantially revised taking into account the comments of all three reviewers. Concerning my comments, the authors have satisfactorily remedied the points I raised and have followed the suggestions I made. I therefore now consider the ms as perfectly suited for publication in Nature Communications